

# Ability of combined soluble urokinase plasminogen activator receptor to predict preventable emergency attendance in older patients in Japan: a prospective pilot study

Toshiya Mitsunaga[1,2], Yuhei Ohtaki[1], Wataru Yajima[1], Kei Sugiura[1], Yutaka Seki[2], Kunihiro Mashiko[2], Masahiko Uzura[1] and Satoshi Takeda[1]

[1] Department of Emergency Medicine, Jikei University School of Medicine, Tokyo, Japan
[2] Department of Emergency Medicine, Association of EISEIKAI Medical and Healthcare Corporation Minamitama Hospital, Tokyo, Japan

Corresponding author
Toshiya Mitsunaga,
toshiya.m@jikei.ac.jp

## ABSTRACT

Soluble urokinase plasminogen activator receptor (suPAR) is a strong and nonspecific inflammatory biomarker that reflects various immunologic reactions, organ damage, and risk of mortality in the general population. Although prior research in acute medical patients showed that an elevation in suPAR is related to intensive care unit admission and risk of readmission and mortality, no studies have focused on the predictive value of suPAR for preventable emergency attendance (PEA). This study aims to evaluate the predictive value of suPAR, which consists of a combination of white blood cell count (WBC), C-reactive protein (CRP), and the National Early Warning Score (NEWS), for PEA in older patients ($>65$ years) without trauma who presented to the emergency department (ED). This single-center prospective pilot study was conducted in the ED of the Association of EISEIKAI Medical and Healthcare Corporation Minamitama Hospital, in Hachiouji City, Tokyo, Japan, from September 16, 2020, to June 21, 2022. The study included all patients without trauma aged 65 years or older who were living in their home or a facility and presented to the ED when medical professionals decided an emergency consultation was required. Discrimination was assessed by plotting the receiver-operating characteristic (ROC) curve and calculating the area under the ROC curve (AUC). During the study period, 49 eligible older patients were included, and thirteen (26.5%) PEA cases were detected. The median suPAR was significantly lower in the PEA group than in the non-PEA group ($p < 0.05$). For suPAR, the AUC for the prediction of PEA was 0.678 (95% CI 0.499–0.842, $p < 0.05$), and there was no significant difference from other variables as follows: 0.801 (95% CI 0.673–0.906, $p < 0.001$) for WBC, 0.833 (95% CI 0.717–0.934, $p < 0.001$) for CRP, and 0.693 (95% CI 0.495–0.862, $p < 0.05$) for NEWS. Furthermore, the AUC for predicting PEA was 0.867 (95% CI 0.741–0.959, $p < 0.001$) for suPAR + WBC + CRP + NEWS, which was significantly higher than that of the original suPAR ($p < 0.01$). The cutoff values, sensitivity, specificity, and odds ratio of suPAR and suPAR + WBC + CRP + NEWS were 7.5 and 22.88, 80.6% and 83.3%, 53.8% and 76.9%, and 4.83 and 16.67, respectively. This study has several limitations. First, this was pilot study, and we included a small number of older patients. Second, the COVID-19 pandemic occurred

during the study period, so that there may be selection bias in the study population. Third, our hospital is a secondary emergency medical institution, and as such, we did not treat very fatal cases, which could be another cause of selection bias. Our single-center study has demonstrated the moderate utility of the combined suPAR as a triage tool for predicting PEA in older patients without trauma receiving home medical care. Before introducing suPAR to the prehospital setting, evidence from multicenter studies is needed.

## INTRODUCTION

Due to improvements in the health care system, life expectancy is increasing in many countries and is currently 81.1 years for men, 87.1 years for women, and 84.2 years for both sexes in Japan (*World Health Statistics, 2019*). The growing aging population is a major theme in public health, and the proportion of people in Japan older than 65 years was 27.7% in 2017 and is expected to reach 31.2% by 2030 (*The Ageing Society, 2018*).

As the population of older adults increases, the number of patients older than 65 years presenting to emergency departments (EDs) is also increasing, and this is one of the major causes of ED overcrowding (*Lee et al., 2018*).

In the past decade, due to an insufficient number of hospital beds and an increase in medical costs, the Japanese government has attempted to shift the care of older patients from in-hospital to at-home care (*Shimizutani Satoshi, 2013*). Many older patients receive home medical care from family physicians or nurses in Japan. When medical professionals recognize a patient in critical condition, they attempt to move that patient to an emergency hospital via ambulance or another form of transportation. According to previous studies (*Parkinson et al., 2020*; *Broek et al., 2020*), 16.1% of emergency patients who attended EDs were nonurgent cases, and 17.5% of older patients admitted to the ED were clinically preventable cases. Although there is a need to safely reduce the number of nonurgent cases of older patients in order to optimize emergency medical care, it is very difficult for medical professionals to perform proper triage of these patients due to several factors, such as frailty, dementia, and atypical physiological reactions.

Several risk-scoring systems have been established to identify the risk of adverse events and death in both EDs and the prehospital setting. The National Early Warning Score (NEWS) was developed in 2012 in the United Kingdom by the National Early Warning Score Development and Implementation Group on behalf of the Royal College of Physicians (*Royal College of Physicians London, 2012*). Our previous study showed that the NEWS had low value for predicting admission and in-hospital mortality in older patients in the prehospital setting and moderate value in the ED setting (*Mitsunaga et al., 2019*). Therefore, a more powerful triage tool for calculating older patients' conditions is needed.

Soluble urokinase plasminogen activator receptor (suPAR) is a strong and nonspecific inflammatory biomarker that reflects various immunologic reactions, organ damage,

and the risk of mortality in the general population (*Hayek et al., 2015*; *Eugen-Olsen et al., 2010*; *Desmedt et al., 2017*; *Rasmussen et al., 2016*). A study conducted on acute medical patients showed that elevation of suPAR is related to intensive care unit admission and risk of readmission and mortality (*Haupt et al., 2012*). Moreover, several studies have demonstrated that combing suPAR with the NEWS or the Danish Emergency Process Triage (DEPT) improved its predictive ability for mortality (*Rasmussen et al., 2018*; *Schultz et al., 2019a*; *Schultz et al., 2019b*). However, most previous suPAR studies were carried out in Denmark (*Schultz et al., 2019a*; *Schultz et al., 2019b*), no study has been conducted in Japan, and the ability of suPAR or suPAR combined with several variables for predicting preventable emergency attendance (PEA) in older emergency patients remains unknown.

The aim of this study is to evaluate the ability of suPAR and combined suPAR to predict PEA in older patients (>65 years) without trauma who present to the ED based on the decision of medical professionals.

## MATERIAL AND METHODS

### Study design

This single-center prospective pilot study was carried out during 21 months in the ED of a secondary emergency institution in Japan to evaluate the ability of suPAR to predict PEA in patients older than 65 years without trauma who lived in their own home or facility and who presented to the ED at the decision of their medical professionals. The protocol for this research project was approved by a suitably constituted Ethics Committee of the institution and conforms to the provision of the Declaration of Helsinki (Committee of Jikei University School of Medicine, approval No. 32-066 (10141)/Committee of Association of EISEIKAI Medical and Healthcare Corporation Minamitama Hospital, approval No. 2020-Ack-04), and written consent was obtained from all the participants.

### Study setting and population

In Japan, EDs are divided into the following three categories: primary emergency institutions see patients with mild conditions; secondary emergency institutions see patients with moderate conditions that require hospitalization; and tertiary emergency institutions resuscitate serious conditions such as multiple trauma, massive bleeding, severe septic shock, and cardiopulmonary arrest (*Tanigawa & Tanaka, 2006*). This study was carried out between September 16, 2020, and June 21, 2022, at the Association of EISEIKAI Medical and Healthcare Corporation Minamitama Hospital, a secondary emergency institution. The hospital is located in Hachiouji City in Tokyo Prefecture. It has 170 beds, and about 5,000 patients present to the ED by ambulance annually. The population of Hachiouji City is 561,622, and 151,845 (27.04%) are older than 65 years (*Hachiouji City Hall, 2020*).

When an emergency call is made to the fire department command center (119), the center provides the information to the emergency medical services nearest to the caller. The emergency medical services rush to the patients by fire department ambulance, gather all patient information on the scene, and then call the proper emergency institution.

In this study, we included all patients aged 65 years or older without trauma who were living in their own home or a facility and who visited the EDs for emergency consultation

based on the decision made by the medical staff. All patients came to the ED by fire department ambulance, elder care and welfare taxi, or their own car.

## Data sources and measurements

We obtained the patients' vital signs and blood samples at the time of their arrival to the ED as soon as possible. suPAR was measured using a suPARnostic® Quick Triage and suPARnostic aLF Quick Test Reader (ViroGates A/S, Birkerød, Denmark).

We also obtained complete blood cell counts, serum chemistry, and blood coagulation measurements using the blood samples.

NEWS was derived from the following seven common physiological vital signs: respiratory rate, peripheral oxygen saturation, presence of inhaled oxygen parameters, body temperature, systolic blood pressure, pulse rate, and Alert, responds to Voice, responds to Pain, Unresponsive score. The scores ranged from 0 to 3 for each parameter. The total NEWS ranged from 0 to 20.

The Charlson Comorbidity Index (CCI) categorizes the comorbidities of patients based on *International Classification of Diseases, Tenth Revision* (ICD-10) diagnostic codes. A weighted score is assigned to each of the 17 comorbidity groups and age, with the scores ranging from 1 to 6. A score of zero indicates no comorbidities (*Haupt et al., 2012*).

The level of care needed in Japan is classified as (1) independence, (2) requiring help 1, (3) requiring help 2, (4) long-term care level 1, (5) long-term care level 2, (6) long-term care level 3, (7) long-term care level 4, and (8) long-term care level 5, requiring the most assistance.

Diagnostic categories were based on ICD-10 and were classified as (1) neurology, (2) pulmonology, (3) cardiology, (4) gastroenterology, (5) endocrinology, (6) nephrology/urology, (7) hematology, (8) collagen disease, (9) gynecology, (10) dermatology, (11) toxicology, and (12) others.

Some patients whose condition appeared to be mild with no abnormality detected on blood tests or imaging were hospitalized in the short term for monitoring. There are three categories of avoidable ED attendances as follows: (1) clinically divertible attendances, (2) clinically preventable attendances, and (3) clinically unnecessary attendances (*Parkinson et al., 2020*). In the present study, medical professionals intervened and determined whether the patients required emergency medical care; thus, we focused on clinically preventable attendances. As mentioned by *Parkinson et al. (2020)*, clinically preventable attendances are defined as preventable emergency admissions. Therefore, based on the study by *Broek et al. (2020)*, we defined PEA as follows: type 1, no somatic causes identified for the patient's initial problem; type 2, no therapeutic or diagnostic interventions planned for the patient's initial problem, except for diagnostics normally conducted in the ED; and type 3, patient's initial problem could have been prevented or avoided by the timely recognition of needs and provision of care before admission (*Broek et al., 2020*).

Patients were followed up until 28 days after their ED visit. We gathered information on the patients' conditions, such as death, by telephone at 48 h, 7 days, and 28 days after emergency consultation.

## Statistical analysis

Continuous variables were described as medians and interquartile ranges and compared using the Student $t$- and Mann–Whitney $U$ tests. Categorical variables were described as numbers and percentages and compared by Fisher's exact test or Pearson's $\chi^2$ test. Receiver-operating characteristic (ROC) analysis and the area under the curve (AUC) were used to evaluate the predictive value of suPAR, white blood cell count (WBC), C-reactive protein (CRP), NEWS, and combined suPAR for the PEA. Confidence intervals (CIs) around the AUC were calculated using bootstrap resampling methods with 1,000 repetitions using R software (R version 3.5.3 binary for OS × 10.11, EI Capitan; *R Core Team, 2019*). We determined the cutoff values as the minimizing the distance from the point in the upper left corner. Using these determined cutoff points, we calculated the sensitivity, specificity, and odds ratios of the suPAR, WBC, CRP, NEWS, and combined suPAR for the prediction of PEA. Calibration was assessed statistically using the Hosmer–Lemeshow C statistic. A statistically significant result suggested a lack of calibration. We performed univariate and multivariate logistic regression analyses to ascertain the effects of factors on the likelihood of PEA. Odds ratios and corresponding 95% CIs were calculated. A $p$-value of <0.05 was considered to indicate statistical significance. A sample size of 49 participants (PEA cases: 13 cases; non-PEA cases: 36 cases) was determined based on 70% power, a 0.05 significance level, allocation ratio of 3, and an expected AUC of 0.7. Although the proportion of PEA cases was about 15%–20% in European countries (*Broek et al., 2020*), we have assumed a PEA rate of 25% because of free access to hospitals in Japan. Data were analyzed by the Statistical Package for the Social Sciences, version 16.0 (SPSS, Chicago, IL, USA).

## RESULTS

During the study period, 49 eligible older patients were included. Table 1 demonstrates the baseline characteristics of this study. The median age (interquartile range) of the patients was 87.0 (9.0) years, and 13 (26.5%) patients were men. The highest level of care needed was long-term care level 5 (11 (22.4%) patients). In terms of visiting route, 24 (49.0%) patients presented to the ED from a facility and 25 (51.0%) from home. Almost all patients came to the ED by fire department ambulance (43 (87.8%) patients). The major comorbidity categories were 37 (75.5%) cardiology cases, 35 (71.4%) neurology cases, and 16 (32.7%) endocrinology cases. The median (interquartile range) CCI of the patients was six (2.0). The major symptoms were 31 (63.3%) fever cases, 26 (53.1%) hypoxia cases, and 16 (32.7%) shortness-of-breath cases. The major diagnostic categories were 14 (28.6%) pulmonology cases, 10 (20.4%) nephrology cases, and eight (16.3%) other cases. Eight (16.3%) patients were discharged from the ED, 38 (77.6%) were admitted to a ward, and three (6.1%) were admitted to the high-care unit. Thirteen (26.5%) PEA cases were detected, of which six (46.2%) with no somatic causes and seven (53.8%) with no planned interventions. Three (6.1%) patients died within 48 h, four (8.2%) died within 7 days, and eight (16.3%) died within 28 days of presenting to the ED. The median (interquartile range) suPAR and NEWS of the patients were 9.0 ng/mL (5.6) and four (4.0), respectively.
**Table 1  Baseline characteristics of the study population.** Data are presented as the median (interquartile range) for continuous variables and number (%) for categorical variables.

| | Total Population ($n$ = 49) Median (interquartile range) |
|---|---|
| Age, years | 87.0 (9.0) |
| Sex [n (%)] | |
|     Male | 13 (26.5) |
|     Female | 36 (73.5) |
| Height, m | 1.5 (0.12) |
| Body weight, kg | 41.0 (15.0) |
| Body mass index: BMI, kg/m2 | 19.1 (3.7) |
| The level of care needed [n (%)] | |
|     Independence | 8 (16.3) |
|     Requiring help 1 | 2 (4.1) |
|     Requiring help 2 | 2 (4.1) |
|     Long-term care level 1 | 5 (10.2) |
|     Long-term care level 2 | 8 (16.3) |
|     Long-term care level 3 | 9 (18.4) |
|     Long-term care level 4 | 4 (8.2) |
|     Long-term care level 5 | 11 (22.4) |
| Visiting route [n (%)] | |
|     Facility | 24 (49.0) |
|     Home | 25 (51.0) |
| Visiting way [n (%)] | |
|     Fire-Department Ambulance | 43 (87.8) |
|     Elder-care and welfare taxi | 3 (6.1) |
|     Own car | 3 (6.1) |
| Comorbidity [n (%)] | |
|     Pulmonology | 8 (16.3) |
|     Cardiology | 37 (75.5) |
|     Neurology | 35 (71.4) |
|     Gastroenterology | 9 (18.4) |
|     Endocrinology | 16 (32.7) |
|     Nephrology | 5 (10.2) |
|     Collagen disease | 1 (2.0) |
|     Cancer | 14 (28.6) |
| Charlson Comorbidity Index | 6 (2.0) |
| Symptoms of chief complain [n (%)] | |
|     Fever | 31 (63.3) |
|     Shock | 9 (18.4) |
|     Hypoxia | 26 (53.1) |
|     Cough | 10 (20.4) |
|     Bloody sputum / Hemoptysis | 2 (4.1) |
|     SOB: Shortness of breath | 16 (32.7) |

**Table 1** (*continued*)

|  |  | Total Population (*n* = 49)<br>Median (interquartile range) |
|---|---|---|
| Stomachache |  | 5 (10.2) |
| Nausea / Vomit |  | 3 (6.1) |
| Bloody stool |  | 1 (2.0) |
| Diarrhea |  | 2 (4.1) |
| Convulsion |  | 1 (2.0) |
| Syncope |  | 1 (2.0) |
| Disturbance of consciousness |  | 12 (24.5) |
| Chest pain |  | 2 (4.1) |
| Fatigue |  | 8 (16.3) |
| Weakness |  | 9 (18.4) |
| Loss of appetite |  | 14 (28.6) |
| Body aches |  | 1 (2.0) |
| Diagnostic category [n (%)] |  |  |
| Pulmonology |  | 14 (28.6) |
| Cardiology |  | 6 (12.2) |
| Neurology |  | 2 (4.1) |
| Gastroenterology |  | 7 (14.3) |
| Endocrinology |  | 2 (4.1) |
| Nephrology |  | 10 (20.4) |
| Others |  | 8 (16.3) |
| Laboratory test |  |  |
| WBC | $[\times 10^3/\mu L]$ | 7.7 (5.7) |
| Hb | [g/dL] | 11.4 (2.9) |
| PLT | $[\times 10^4/\mu L]$ | 20.6 (9.0) |
| TP | [g/dL] | 6.8 (1.1) |
| Alb | [g/dL] | 3.2 (0.7) |
| AST | [U/L] | 24.0 (21.0) |
| ALT | [U/L] | 17.0 (19.0) |
| LDH | [U/L] | 240.0 (127.0) |
| ChE | [U/L] | 186.0 (99.0) |
| CK | [U/L] | 58.0 (89.0) |
| T-Bil | [mg/dL] | 0.8 (0.4) |
| ALP | [U/L] | 268.0 (154.0) |
| $\gamma$-GT | [U/L] | 21.0 (24.0) |
| BUN | [mg/dL] | 24.0 (19.0) |
| Cr | [mg/dL] | 0.88 (0.49) |
| Na | [mEq/L] | 138.0 (7.0) |
| K | [mEq/L] | 4.3 (0.8) |
| Cl | [mEq/L] | 101.0 (8.0) |
| CRP | [mg/dL] | 3.22 (9.9) |
| PT | [%] | 79.6 (19.8) |
| APTT | [sec] | 32.1 (6.8) |

**Table 1** (*continued*)

| | | Total Population ($n = 49$) Median (interquartile range) |
|---|---|---|
| Fib | [mg/dL] | 334.0 (134.0) |
| Disposition [n (%)] | | |
| Discharge | | 8 (16.3) |
| Admission to a ward | | 38 (77.6) |
| Admission to High Care Unit | | 3 (6.1) |
| PEA: Preventable Emergency Attendance [n (%)] | | 13 (26.5) |
| The categories of PEA [n (%)] | | |
| Type 1 | | 6 (46.2) |
| Type 2 | | 7 (53.8) |
| Type 3 | | 0 (0) |
| Mortality after Emergency Department visit [n (%)] | | |
| 48 h | | 3 (6.1) |
| 7 days | | 4 (8.2) |
| 28 days | | 8 (16.3) |
| suPAR [ng/mL] | | 9.0 (5.6) |
| NEWS | | 4 (4.0) |

**Notes.**

WBC, white blood cell count; Hb, hemoglobin; PLT, platelets; TP, total protein; Alb, albumin; AST, aspartate aminotransferase; ALT, alanine aminotransferase; LDH, lactate dehydrogenase; ChE, cholinesterase; CK, creatinine phosphokinase; T-Bil, total bilirubin; ALP, alkaline phosphatase; $\gamma$-GT, $\gamma$-glutamyl transpeptidase; BUN, blood urea nitrogen; Cr, creatinine; Na, sodium; K, potassium; Cl, chlorine; CRP, C-reactive protein; PT, prothrombin time; APTT, activated partial thromboplastin time; Fig, fibrinogen; suPAR, soluble urokinase plasminogen activator receptor; NEWS, National Early Warning Score.

Table 2 shows the comparison of parameters between the PEA and non-PEA groups. There was no significant difference between the PEA and non-PEA groups regarding age, sex, body mass index, level of care needed, visiting route, comorbidity, CCI, or symptoms. The proportion of patients with pulmonary disease was significantly higher in the non-PEA group than in the PEA group ($p < 0.05$). Inflammatory markers such as WBC and CRP were significantly lower in the PEA group than in the non-PEA group ($p < 0.01$ and $p < 0.001$, respectively); in addition, the albumin level was significantly higher in the PEA group than in the non-PEA group ($p < 0.05$). suPAR and NEWS were significantly lower in the PEA group than in the non-PEA group ($p < 0.05$ and $p < 0.05$, respectively).

Table 3 shows the results of the univariate and multivariate logistic regression analysis of the factors associated with PEA. We found that low WBC ($<8.0 \times 10^3/\mu L$) and CRP ($<1.0$ mg/dL) were significant predictors of PEA in the univariate logistic regression analysis, and their odds ratios were 8.64 and 5.60, respectively. In addition, multivariate logistic regression analysis showed that low WBC ($<8.0 \times 10^3/\mu L$) was a significant predictor for PEA, with an odds ratio of 6.28.

Table 4 displays the ROC analysis and Hosmer–Lemeshow fit test for the prediction of PEA. The AUC of suPAR was the same as that of NEWS and lower, albeit not significantly, than that of the WBC and CRP level. The AUC for predicting PEA was 0.678 (95% CI [0.499–0.842], $p < 0.05$) for suPAR, 0.693 (95% CI [0.495–0.862], $p < 0.05$) for NEWS, 0.801 (95% CI [0.673–0.906], $p < 0.001$) for WBC, and 0.833 (95% CI [0.717–0.934], $p < 0.001$) for CRP.
**Table 2 Comparison of parameters between the PEA and non-PEA groups.** Data are presented as the median (interquartile range) for continuous variables and number (%) for categorical variables.

| | Median (interquartile range) | | |
|---|---|---|---|
| | PEA (*n* = 13) | Non-PEA (*n* = 36) | *p* value |
| Age, years | 90.0 (12.0) | 87.0 (8.3) | 0.61 |
| Sex [n (%)] | | | |
| Male | 2 (15.4) | 11 (30.6) | 0.47 |
| Female | 11 (84.6) | 25 (69.4) | |
| Height, m | 1.5 (0.08) | 1.5 (0.14) | 0.70 |
| Body weight, kg | 40.0 (14.0) | 41.5 (14.0) | 0.69 |
| Body Mass Index: BMI, kg/m2 | 18.9 (3.4) | 19.2 (3.8) | 0.84 |
| The level of care needed [n (%)] | | | |
| Independence | 1 (7.7) | 7 (19.4) | |
| Requiring help 1 | 0 (0.0) | 2 (5.6) | |
| Requiring help 2 | 1 (7.7) | 1 (2.8) | |
| Long-term care level 1 | 1 (7.7) | 4 (11.1) | 0.75 |
| Long-term care level 2 | 3 (23.1) | 5 (13.9) | |
| Long-term care level 3 | 3 (23.1) | 6 (16.7) | |
| Long-term care level 4 | 0 (0.0) | 4 (11.1) | |
| Long-term care level 5 | 4 (30.8) | 7 (19.4) | |
| Visiting route [n (%)] | | | |
| Facility | 5 (38.5) | 19 (52.8) | 0.38 |
| Home | 8 (61.5) | 17 (47.2) | |
| Comorbidity [n (%)] | | | |
| Pulmonology | 2 (15.4) | 6 (16.7) | |
| Cardiology | 10 (76.9) | 27 (75.0) | |
| Neurology | 11 (84.6) | 24 (66.7) | |
| Gastroenterology | 3 (23.1) | 6 (16.7) | 0.88 |
| Endocrinology | 6 (46.2) | 10 (27.8) | |
| Nephrology | 2 (15.4) | 3 (8.3) | |
| Collagen disease | 0 (0.0) | 1 (2.8) | |
| Cancer | 2 (15.4) | 12 (33.3) | |
| Charlson Comorbidity Index | 5.0 (1.0) | 6.0 (2.0) | 0.13 |
| Symptoms of chief complain [n (%)] | | | |
| Fever | 4 (30.8) | 27 (75.0) | |
| Shock | 1 (7.7) | 8 (22.2) | |
| Hypoxia | 2 (15.4) | 24 (66.7) | |
| Cough | 0 (0.0) | 10 (27.8) | |
| SOB: Shortness of breath | 3 (23.1) | 13 (36.1) | |
| Stomachache | 2 (15.4) | 3 (8.3) | |
| Nausea / Vomit | 0 (0.0) | 3 (8.3) | |
| Bloody stool | 0 (0.0) | 1 (2.8) | |
| Diarrhea | 0 (0.0) | 2 (5.6) | |

**Table 2** (*continued*)

| | | Median (interquartile range) | | 0.36 |
|---|---|---|---|---|
| | | PEA (*n* = 13) | Non-PEA (*n* = 36) | *p* value |
| Convulsion | | 0 (0.0) | 1 (2.8) | |
| Syncope | | 1 (7.7) | 0 (0.0) | |
| Disturbance of consciousness | | 4 (30.8) | 8 (22.2) | |
| Chest pain | | 0 (0.0) | 2 (5.6) | |
| Fatigue | | 1 (7.7) | 7 (19.4) | |
| Weakness | | 1 (7.7) | 8 (22.2) | |
| Loss of appetite | | 4 (30.8) | 10 (27.8) | |
| Body aches | | 0 (0.0) | 1 (2.8) | |
| Diagnostic category [n (%)] | | | | |
| Pulmonology | | 1 (7.7) | 13 (36.1) | |
| Cardiology | | 1 (7.7) | 5 (13.9) | |
| Neurology | | 1 (7.7) | 1 (2.8) | |
| Gastroenterology | | 2 (15.4) | 5 (13.9) | <0.05 |
| Endocrinology | | 0 (0.0) | 2 (5.6) | |
| Nephrology | | 2 (15.4) | 8 (22.2) | |
| Others | | 6 (46.2) | 2 (5.6) | |
| Laboratory test | | | | |
| WBC | [$\times 10^3/\mu$L] | 6.2 (3.0) | 10.3 (6.7) | <0.01 |
| Hb | [g/dL] | 11.3 (2.0) | 12.0 (3.5) | 0.74 |
| PLT | [$\times 10^4/\mu$L] | 19.0 (4.8) | 21.0 (11.0) | 0.30 |
| TP | [g/dL] | 6.7 (1.0) | 6.8 (1.1) | 0.64 |
| Alb | [g/dL] | 3.5 (0.6) | 3.1 (0.7) | <0.05 |
| AST | [U/L] | 27.0 (15.0) | 24.0 (22.0) | 0.22 |
| ALT | [U/L] | 13.0 (22.0) | 19.0 (13.8) | 0.56 |
| LDH | [U/L] | 213.0 (88.0) | 249.0 (151.0) | 0.15 |
| ChE | [U/L] | 240.0 (86.0) | 173.5 (82.3) | 0.05 |
| CK | [U/L] | 47.0 (42.0) | 70.5 (103.5) | 0.61 |
| T-Bil | [mg/dL] | 0.7 (0.5) | 0.8 (0.4) | 0.40 |
| ALP | [U/L] | 241.0 (47.0) | 294.0 (203.0) | 0.05 |
| $\gamma$-GT | [U/L] | 21.0 (26.0) | 21.0 (21.0) | 0.70 |
| BUN | [mg/dL] | 19.0 (7.0) | 27.0 (23.0) | <0.05 |
| Cr | [mg/dL] | 0.77 (0.32) | 0.93 (0.60) | <0.05 |
| Na | [mEq/L] | 140.0 (3.0) | 137.0 (7.3) | 0.25 |
| K | [mEq/L] | 4.0 (1.1) | 4.3 (0.7) | 0.17 |
| Cl | [mEq/L] | 102.0 (6.0) | 100.0 (8.0) | 0.15 |
| CRP | [mg/dL] | 0.87 (1.18) | 5.9 (9.4) | <0.001 |
| PT | [%] | 83.8 (14.7) | 78.0 (22.1) | 0.23 |
| APTT | [sec] | 29.1 (5.9) | 32.9 (6.6) | 0.06 |
| Fib | [mg/dL] | 311.0 (110.0) | 338.5 (144.0) | 0.37 |
| Disposition [n (%)] | | | | |
| Discharge | | 8 (61.5) | 0.0 (0.0) | |

| | Median (interquartile range) | | |
|---|---|---|---|
| | **PEA (*n* = 13)** | **Non-PEA (*n* = 36)** | ***p* value** |
| | | | <0.001 |
| Admission to a ward | 5 (38.5) | 33 (91.7) | |
| Admission to High Care Unit | 0 (0.0) | 3 (8.3) | |
| suPAR [ng/mL] | 7.4 (4.4) | 9.3 (5.4) | <0.05 |
| NEWS | 3.0 (5.0) | 5.0 (4.3) | <0.05 |

**Notes.**
WBC, white blood cell count; Hb, hemoglobin; PLT, platelets; TP, total protein; Alb, albumin; AST, aspartate aminotransferase; ALT, alanine aminotransferase; LDH, lactate dehydrogenase; ChE, cholinesterase; CK, creatinine phosphokinase; T-Bil, total bilirubin; ALP, alkaline phosphatase; $\gamma$-GT, $\gamma$-glutamyl transpeptidase; BUN, blood urea nitrogen; Cr, creatinine; Na, sodium; K, potassium; Cl, chlorine; CRP, C-reactive protein; PT, prothrombin time; APTT, activated partial thromboplastin time; Fig, fibrinogen; suPAR, soluble urokinase plasminogen activator receptor; NEWS, National Early Warning Score.

**Table 3** Univariate and multivariate logistic regression analysis of factors associated with PEA.

| Predictor | Univariate | | Multivariate | |
|---|---|---|---|---|
| | **Odds ratio (95% CI)** | ***p* value** | **Odds ratio (95% CI)** | ***p* value** |
| Age $\geq$ 90 years (vs <90 years) | 1.83 (0.51–6.59) | 0.35 | | |
| Sex | | | | |
| Female (vs male) | 2.42 (0.46–12.79) | 0.30 | | |
| Visiting route | | | | |
| Home (vs Facility) | 1.79 (0.49–6.53) | 0.38 | | |
| Laboratory test | | | | |
| WBC $\leq$ 8,000 (vs >8,000) | 8.64 (1.66–44.95) | <0.05 | 6.28 (1.14–34.64) | <0.05 |
| CRP $\leq$ 1.0 mg/dL (vs >1.0) | 5.60 (1.43–21.95) | <0.05 | 3.69 (0.85–15.97) | 0.08 |
| Alb $\geq$ 3.5 g/dL (vs <3.5) | 3.03 (0.82–11.26) | 0.10 | | |
| suPAR $\leq$ 7.5 ng/mL (vs >7.5) | 3.50 (0.93–13.18) | 0.06 | | |
| NEWS $\leq$ 3.0 (vs >3.0) | 3.50 (0.93–13.18) | 0.06 | | |

**Notes.**
WBC, white blood cell count; CRP, C-reactive protein; Alb, albumin; suPAR, soluble urokinase plasminogen activator receptor; NEWS, National Early Warning Score.

Combining other factors with suPAR improved the predictive value for PEA, and the AUC for predicting PEA was 0.797 (95% CI [0.656–0.922], $p < 0.001$) for suPAR + WBC, 0.822 (95% CI [0.693–0.926], $p < 0.001$) for suPAR + CRP, 0.736 (95% CI [0.579–0.889], $p < 0.01$) for suPAR + NEWS, 0.860 (95% CI [0.752–0.957], $p < 0.001$) for suPAR + WBC + CRP, 0.807 (95% CI [0.665–0.926], $p < 0.001$) for suPAR + WBC + NEWS, 0.829 (95% CI [0.694–0.936], $p < 0.001$) for suPAR + CRP + NEWS, and 0.867 (95% CI [0.741–0.959], $p < 0.001$) for suPAR + WBC + CRP + NEWS. Almost all of the AUCs of the suPAR scores combined with other factors were significantly greater than that of the original suPAR.

The cutoff values for PEA were 7.4 for WBC, 2.04 for CRP, 4.0 for NEWS, 7.5 for suPAR, 18 for suPAR + WBC, 12.98 for suPAR + CRP, 13.8 for suPAR + NEWS, 22.97 for suPAR + WBC + CRP, 20.7 for suPAR + WBC + NEWS, 16.98 for suPAR + CRP + NEWS, and 22.88 for +WBC + CRP + NEWS, respectively. All scores were well calibrated for predicting PEA.

**Table 4  Receiver-operating characteristic curve analysis and HosmerLemeshow fit test for the prediction of PEA.**

| | | Area Under the Curve (95% CI) | $p$-value | Cut-off values | Sensitivity | Specificity | Odds ratio | Hosmer-Lemeshow C statistic (Chi-Square) |
|---|---|---|---|---|---|---|---|---|
| 1 | WBC | 0.801 (0.673–0.906) | $p < 0.001$ | 7.4 | 69.4% | 76.9% | 7.58 | 4.161 |
| 2 | CRP | 0.833 (0.717–0.934) | $p < 0.001$ | 2.04 | 75.0% | 92.3% | 36.00 | 5.012 |
| 3 | NEWS | 0.693 (0.495–0.862) | $p < 0.05$ | 4.0 | 75.0% | 53.8% | 3.50 | 8.392 |
| 4 | suPAR | 0.678 (0.499–0.842) | $p < 0.05$ | 7.5 | 80.6% | 53.8% | 4.83 | 3.292 |
| 5 | suPAR+WBC | 0.797 (0.656–0.922) | $p < 0.001$ | 18 | 69.4% | 76.9% | 7.58 | 5.681 |
| 6 | suPAR+CRP | 0.822 (0.693–0.926) | $p < 0.001$ | 12.98 | 69.4% | 84.6% | 12.50 | 2.300 |
| 7 | suPAR+NEWS | 0.736 (0.579–0.889) | $p < 0.01$ | 13.8 | 63.9% | 76.9% | 5.90 | 13.365 |
| 8 | suPAR+WBC+CRP | 0.860 (0.752–0.957) | $p < 0.001$ | 22.97 | 69.4% | 92.3% | 27.27 | 3.563 |
| 9 | suPAR+WBC+NEWS | 0.807 (0.665–0.926) | $p < 0.001$ | 20.7 | 72.2% | 76.9% | 8.67 | 7.084 |
| 10 | suPAR+CRP+NEWS | 0.829 (0.694–0.936) | $p < 0.001$ | 16.98 | 75.0% | 84.6% | 16.50 | 5.454 |
| 11 | suPAR+WBC+CRP+NEWS | 0.867 (0.741–0.959) | $p < 0.001$ | 22.88 | 83.3% | 76.9% | 16.67 | 4.706 |

**Notes.**

WBC, white blood cell count; CRP, C-reactive protein; suPAR, soluble urokinase plasminogen activator receptor; NEWS, National Early Warning Score.

$p < 0.05$: 3 vs 11, 4 vs 9, 4 vs 10, 5 vs 8, 7 vs 10, 7 vs 11, 9 vs 11. $p < 0.01$: 4 vs 5, 4 vs 6, 4 vs 8, 4 vs 11.

# DISCUSSION

Our data demonstrated that suPAR was significantly higher in the non-PEA group; moreover, the present study also showed the moderate utility of suPAR combined with WBC, CRP, and NEWS for predicting PEA. This is the first study to evaluate the usefulness of suPAR when combined with WBC, CRP, and NEWS in EDs for predicting PEA in the older population in Japan.

This was a prospective pilot study. We determined the sample size with a relatively low power of 70% and an AUC of 0.7. In studies carried by *Rasmussen et al. (2016)*, *Schultz et al. (2019a)*, and *Schultz et al. (2019b)*, the AUC of suPAR for in-hospital mortality ranged from 0.84 to 0.92, which was almost the same AUC reported in previous studies for in-hospital mortality (0.894–0.902) for NEWS (*Smith et al., 2013*; *Kovacs et al., 2016*). In the study conducted by *Sbiti-Rohr et al. (2016)*, the AUC of NEWS for intensive care unit admission was 0.73, although no study has calculated the AUC of suPAR for admission or PEA in the older population. As reported previously in this study, the AUC of suPAR for in-hospital mortality was almost the same as that of NEWS; thus, we hypothesized that the AUC of suPAR for PEA would be equivalent to the AUC of NEWS for admission in the previous study, and we determined the AUC of the sample size for PEA to be 0.7. In this study, we included 13 cases of PEA and 36 of non-PEA based on the sample size test, and the AUC of suPAR for PEA was 0.678, which was almost the same as the set value of 0.7; thus, the original sample test may be valid.

In our study, an almost similar number of cases of PEA with type 1 and 2 causes were identified, whereas no cases of PEA with type 3 causes were identified. In more than 80% of the clinics that provide home-visit medical care, there is only one physician, and it is often difficult to handle home-based patients in parallel with regular medical care. Therefore,

it is expected that a sufficient medical examination cannot be provided, and the severity of the patient may be determined based on symptoms (*Japan Medical Association Research Institute, 2016*).

According to several studies (*Laou et al., 2022*; *Lin et al., 2021*), the suPAR level may change depend on the type of trauma, so that we did not include the patients with trauma in this study.

In their study, *Broek et al. (2020)* found the factors associated with PEA to be older age and a low urgency classification; however, there has not been enough research evaluating the factors related to PEA in older patients. In the comparison parameters between the PEA and non-PEA groups, we found no difference between the groups regarding the level of care required, visiting route, comorbidity, or CCI. Therefore, the patients' basic condition may not be associated with PEA. The proportion of pulmonary symptoms such as hypoxia, cough, and shortness of breath was higher in the non-PEA group; as a result, the proportion of pulmonary and cardiac diagnoses was significantly higher in the non-PEA group. The study conducted by *Nojiri et al. (2019)* demonstrated that the incidence of pneumonia and coronary heart disease has recently increased in both males and females in Japan. Patients with pulmonary or cardiac diseases often require oxygenation and thus may be hospitalized more often than those with other diagnostic categories. On the other hand, the proportion of disturbance in consciousness was higher in the PEA group, but most cases had no specific somatic causes.

A study carried by *Rafal et al. (2020)* revealed that the suPAR level of elderly people was higher than that of younger people. Moreover, the suPAR level of advanced elderly people older than 79 years was even higher with increasing age. In our study, there was no significant difference of age between the PEA group and the non-PEA group, so that we assumed that the age distribution between the two groups had no effect on the suPAR level.

The present study also demonstrated that low WBC was significantly associated with PEA, whereas older age, low suPAR, and low NEWS were not significantly associated with PEA. As reported in a previous study, the patients in the PEA group in our study were older, whereas the median age was very high, and the number of PEAs was small; thus, our data did not demonstrate a significant association between older age and PEA.

We calculated the ability of suPAR to predict PEA in older emergency patients, but no excellent (AUC >0.90) or good (AUC >0.8) ability was found. As with NEWS, the present study showed that suPAR had relatively low effectiveness for predicting PEA in the older population, and the AUC for PEA was 0.678. There was no significant difference between the AUC of suPAR, NEWS, WBC, and CRP for predicting PEA.

According to a previous study carried out by *Rasmussen et al. (2018)*, the AUC of suPAR combined with age, sex, and NEWS for predicting in-hospital mortality was 0.92, which is significantly greater than that of suPAR alone (0.84, $p < 0.0001$). In their study, *Schultz et al. (2019a)* and *Schultz et al. (2019b)* showed that the AUC of suPAR combined with DEPT for all-cause mortality was higher than that of the original suPAR, but the difference was not significant (0.87 *vs* 0.85, $p = 0.16$). However, no other study has evaluated the predictive ability of suPAR combined with other variables for mortality or PEA. In our

study, age was not found to be a significant factor associated with PEA, whereas WBC, CRP, and NEWS were. Therefore, we combined WBC, CRP, and NEWS with suPAR, but not age and suPAR. Furthermore, the AUC of suPAR with WBC, CRP, and NEWS was significantly higher than that of the original suPAR for predicting PEA (0.678 $vs$ 0.867, $p < 0.01$), and this AUC value was moderate and determined to be a good predictor of PEA.

Our study showed that a suPAR cutoff value of 7.5 can safely reduce the emergency visits of patients in the PEA group by 61.5%, except for potentially mortal patients, although 19.4% of the patients who should be hospitalized might be inadequately triaged. On the other hand, the cutoff value of 22.88 for suPAR + WBC + CRP + NEWS can safely reduce the emergency visits of patients in the PEA group by 76.9%, except for potentially mortal patients, although 16.7% of the patients who should be hospitalized may be inadequately triaged. Although we conducted our study in the ED setting, all cases received an intervention by medical professionals, and our results show that in the prehospital setting, the combined suPAR may safely reduce the number of PEA cases more than the original suPAR. Moreover, the WBC or CRP results can be obtained in a short time, just like NEWS, so that prehospital medical professionals can easily calculate the combined score and also predict PEA cases.

This study has several limitations. First, this was pilot study, and we included a small number of older patients. Thus, a statistically significant difference was difficult to obtain between the PEA and non-PEA groups. Second, the COVID-19 pandemic occurred during the study period, and our hospital frequently had to stop accepting emergency cases; thus, there may be selection bias in the study population. Third, our hospital is a secondary emergency medical institution, and as such, we did not treat very fatal cases, which could be another cause of selection bias. Further multicenter, large-population studies are needed for external validation of our results.

## CONCLUSIONS

Our single-center study has shown the moderate utility of the combined suPAR for predicting PEA and the low utility of the original suPAR for predicting PEA in older patients without trauma who received home medical care. To introduce the combined suPAR in the prehospital setting, evidence from multicenter studies is needed.

### Funding
This work was supported by JSPS KAKENHI Grant Number JP20K17910. The funders had no role in study design, data collection and analysis, decision to publish, or preparation of the manuscript.

### Grant Disclosures
The following grant information was disclosed by the authors:
JSPS KAKENHI: JP20K17910.

## Competing Interests

The authors declare there are no competing interests.

## Author Contributions

- Toshiya Mitsunaga conceived and designed the experiments, prepared figures and/or tables, authored or reviewed drafts of the article, and approved the final draft.
- Yuhei Ohtaki performed the experiments, prepared figures and/or tables, and approved the final draft.
- Wataru Yajima performed the experiments, analyzed the data, prepared figures and/or tables, and approved the final draft.
- Kei Sugiura performed the experiments, analyzed the data, prepared figures and/or tables, and approved the final draft.
- Yutaka Seki conceived and designed the experiments, authored or reviewed drafts of the article, and approved the final draft.
- Kunihiro Mashiko conceived and designed the experiments, authored or reviewed drafts of the article, and approved the final draft.
- Masahiko Uzura conceived and designed the experiments, authored or reviewed drafts of the article, and approved the final draft.
- Satoshi Takeda conceived and designed the experiments, authored or reviewed drafts of the article, and approved the final draft.

## Human Ethics

The following information was supplied relating to ethical approvals (*i.e.*, approving body and any reference numbers):

Committee of Jikei University School of Medicine, approval No. 32-066 [10141]/Committee of Association of EISEIKAI Medical and Healthcare Corporation Minamitama Hospital, approval No. 2020-Ack-04.

## Data Availability

The raw data are available in the Supplementary File.

## Supplemental Information

Supplemental information for this article can be found online at http://dx.doi.org/10.7717/peerj.14322#supplemental-information.

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
