# Peer review of "Ability of combined soluble urokinase plasminogen activator receptor to predict preventable emergency attendance in older patients in Japan: a prospective pilot study"

_PeerJ, doi:10.7717/peerj.14322_

## Round 0.1 · original submission · Major Revisions

The reviewers have indicated several important points that you have to consider. Note that the revised paper will be sent back to the original reviewers and I cannot guarantee it's acceptance.

Reviewer 1 ·

Basic reporting

1) Clear and unambiguous, professional English used throughout - Yes

2) Literature references, sufficient field background/context provided - Yes

3) Professional article structure, figures, tables. Raw data shared - Yes

4) Self-contained with relevant results to hypotheses - Yes

Experimental design

1) Original primary research within Aims and Scope of the journal
- Within the Scope of the Journal - Yes.
- Original primary research: Partially
Authors mentions couple of several studies being done previously in discussion (Line # 294-298 and Line # 337-340), in my opinion, the authors have tried to reproduce some of the data and included their point of view of considering the levels of suPAR in assessing the PEA.

2) Research question well defined, relevant & meaningful. It is stated how research fills an identified knowledge gap - Yes.

3) Rigorous investigation performed to a high technical & ethical standard - Partially.
The authors mention a couple of pitfalls of their manuscripts in the discussion (Line # 360 - 366). It is very appreciative of the authors for being honest, simultaneously, it also suggest that there were limited attempts made for any rigorous investigations.

4) Methods described with sufficient detail & information to replicate.

Validity of the findings

1) All underlying data have been provided; they are robust, statistically sound, & controlled - Yes

2) Conclusions are well stated, linked to original research question & limited to supporting results - Yes.

Additional comments

1) The data is collected from old age patients, which may be going through some other pathological conditions. Is is possible that those pathological condition and its subsequent treatment may have an effect on suPAR levels. It is highly recommended to provide some insights on the basal suPAR levels in the Comorbidities mentioned in Table-1. In addition to that does the suPAR levels changes in presence or absence of trauma? This information may help to strengthen the point of selecting the patients not going under trauma.

2) Referring to data from Tabel-2
As a generalized statement, higher levels of albumin (protein) may indicate a damaged or a pathological condition like inflammation. The authors mentioned that suPAR is one of the inflammatory marker and so is CRP. It is interesting that the levels of two inflammatory markers are decreased, but albumin levels are significantly upregulated. Also the numerical values for albumin in PEA versus non-PEA is 3.5 and 3.1, with a difference of 0.4 g/dL. It is very strange that such a small difference is significant. The authors should mention either standard error of mean or standard deviation and which statistical test has been performed to determine significance.

3) It is recommended to perform co-relation studies with the present data or with additional data from multi-center large population studies between the parameters selected.

Reviewer 2 ·

Basic reporting

The article is ambiguous. Though its English is generally understandeble, the vocabulary and expression should be changed to make it more professional. There are sentences that are not clear enough and therefore the idea of the study (together with statistical analysis and its interpretation) is not clear. The article provides basic literature references on the suPAR, nevertheless more deep research should be done - there are articles considering suPAR in elderly (i.e. doi: 10.1038/s41598-020-72377-w.), authors should at least discuss it and check if it corresponds anyhow to their studies. The structure of the article is professional, although the format of information in tables are missing (i e. the unit of suPAR concentration).
Also statistical relevances should be provided in details. For example: p<0,05 in ROC analysis of suPAR and NEWS should be datailed, since the 95CI starts below 0,5 in both the markers.

Experimental design

The idea of the research is - to the reviewr's knowledge - oryginal. Although the research question is defined it is hard to understand the final conclusions in correspondence with the results and statistical analysis. The methods are described in details, even in to much details - there is no need to provide step-by-step protocol of the suPAR measurement, because it is written down it the reagent kit. Instead of that the methodology of performance of the rest laboratory parameters included in the study should be provided. Some information, like the way of how the patients got to the hospital also seems unnecessary.

Validity of the findings

The most important findings that should correspond to the aim of the study are missing among the huge amount of details provided. The statistical analysis remains not clear. It is not clear why and how authors calculated the AUC of suPAR for prediction of PED - in relation to what patients? And why there are two ROC analysis - another one for the non-PED patients, while the article suggests that only PED and non-PED patiens are the studied population. How did the authors calulate the "combined suPAR" ? The cutof provided in the ROC analysis suggests that it is a simple sum of the numbers (suPAR+WBC+CRP+NEWS numerical tests results), which is mathematically wrong. And the statistical analysis seems controvertial for such small number of patients (even for pilot study). Authors stated that there are aprox 5000 patients annually admitted to the ED - so even during the pandemy it should be more than just 48 out of potentially 10000 patients.

The conclusions are unclear - why authors assess the prediction of PED if majority of results numbers are higher for non-PED? Or maybe the hypothesis should be changed?

Additional comments

The article requires major and thorough revision

---

## Round 0.2 · Minor Revisions

Thank you for revising the manuscript according to the reviewers suggestions. Please, consider to include in the abstract the most important outcomes with the CI, instead of using percentage only. Also include the number of subjects included in the study and sentence stating the limitations of the study in the abstract.

---

## Round 0.3 · accepted · Accept

Thank you for revising again your manuscript and for improving the abstract of the revised manuscript. I am glad to accept your manuscript.